# Can Multidisciplinary Inpatient and Outpatient Rehabilitation Provide Sufficient Prevention of Disability in Patients with a Brain Tumor?—A Case-Series Report of Two Programs and A Prospective, Observational Clinical Trial

**DOI:** 10.3390/ijerph17186488

**Published:** 2020-09-06

**Authors:** Katarzyna Hojan, Karolina Gerreth

**Affiliations:** 1Department of Rehabilitation in the Greater Poland Cancer Centre, 15 Garbary St. 61-866 Poznan, Poland; 2Neurorehabilitation Ward, Provincial Hospital in Poznan, 9-14 Juraszow St. 60-479 Poznan, Poland; 3Department of Risk Group Dentistry, Chair of Pediatric Dentistry, Poznan University of Medical Sciences, 70 Bukowska Street, 60-812 Poznan, Poland; karolinagerreth@poczta.onet.pl

**Keywords:** oncology, exercise, supportive care, cancer, quality of life

## Abstract

Brain tumor (BT) patients have a high incidence of disability due to the effects of the tumor itself or oncological treatment. Despite the incidence of neurological and functional deficits caused by BT, rehabilitation of those patients is not as properly established as in patients with other neurological conditions. The aim of the research was to evaluate the effectiveness of a multidisciplinary rehabilitation, carried out as an out- or in-patient program, as prevention of disability in BT patients. This was developed as a case-series report of two programs and a prospective, observational clinical study in BT patients who were allocated to inpatient (n = 28) or outpatient (n = 26) rehabilitation programs. The patients were assessed using the Barthel Index, Berg Balance Scale, Functional Independence Measure (FIM), Functional Assessment of Cancer Therapy—Brain and Cognitive Function, and Addenbrooke’s Cognitive Examination III (ACE III) upon admission and after 12 weeks of rehabilitation. Analysis of the results showed that patients in both programs significantly improved their physical functioning scores in daily activities (*p* < 0.0001). We also observed significant reductions in most post-intervention cognitive complaints (*p* < 0.05), except for the FIM social functioning and ACE III language functioning in the outpatient group (*p* > 0.05) in contrast to inpatients (*p* < 0.001). This is evidence that early multidisciplinary rehabilitation is an effective therapeutic strategy to reduce BT symptoms and disability in this group of patients.

## 1. Introduction

Globally, a total of 296,851 new cases (age-standardized rate, ASR = 3.9 and 3.1, male and female, respectively) of brain and other central nervous system (CNS) cancers were diagnosed in 2018 [1]. The most common histological type of primary CNS cancer is glioma as a group of malignant brain tumors (BT), including high-grade glioma or glioblastoma and low-grade gliomas (astrocytoma, oligodendroglioma), and other tumors of glial origin (ependymomas, schwannomas), medulloblastomas, CNS lymphomas, and meningiomas [2,3]. Due to clinical and technological advancements in oncological therapy, the survival rate of patients with primary BT has significantly increased in recent years [1,4], generating a growing need for rehabilitation treatment to diminish neurological impairments and restore their quality of life (QoL) [5,6]. Disability in this group of cancer patients may be due to the dysfunction of the CNS, depending on the location and size of the removed tumor, the method of oncological treatment (surgery, chemotherapy, radiotherapy), and the following cancer therapy [3,6]. Thus, patients with BT and their families need to be aware of the consequences of disability effects caused by the tumor itself or the side effects of oncological treatment [5]. BT patients present significant functional and psychosocial impairments that limit their participation in daily activities. Many of those are connected with neurological deficits, leading to behavioral, cognitive, and physical dysfunctions [3]. In addition, BTs and their onco-therapy can have a negative impact on patients’ QoL including their family life, work, and self-sufficiency, which is associated with significant costs and socioeconomic implications like increased requirements for health care, and social care services [5,6,7,8]. As these difficulties have complex manifestations, multidisciplinary expertise is necessary to evaluate the influence of each variable in the rehabilitation plan in order to provide appropriate support. Contemporary evidence for the effectiveness of rehabilitation treatment is favorable in many cancer patients [5,9]. Despite the high probability of clinical complications that result in disabilities, dysfunction, or neurological symptoms, functional and cognitive rehabilitation following cancer treatment in BT patients is still fairly uncommon [5,9]. Literature data [9] confirm that BT patients were interested in physical exercise and felt that they would be able to exercise at home. Home-based rehabilitation may be particularly important for those BT patients who additionally have neurological symptoms (e.g., epilepsy) which may restrict their mobility [10]. Some pilot studies with small groups of patients [11,12] acknowledge physical exercise therapy for BT patients as feasible and safe if relevant inclusion criteria and precautionary screening are followed [12]. However, the application of comprehensive rehabilitation in patients after BT treatment in terms of the assessment of disability and daily functioning in this group of patients has not been sufficiently documented. A longitudinal study of BT patients living in the community after treatment (median time since diagnosis 2.1 years) reported ataxia (44%), seizures (43%), paresis (37%), cognitive dysfunction (36%), and visual impairment (35%) [13]. An analysis of rehabilitation services for BT patients after the first year of oncological treatment has shown an insufficient percentage of rehabilitation interventions in this population [14] and indicated the need to conduct further research concerning appropriate multidisciplinary rehabilitation [13]. Other authors [6,13] have noted a lack of recommendations for rehabilitation management in BT patients. There are still no references as to the duration and type of rehabilitation that will be well-tolerated by this group, especially due to the fact that these are patients with a heterogeneous disease course and improvements are unlikely to be universal or guaranteed by participation in a rehabilitation program. 

Therefore, the aim of the present study was a clinical observation concerning assessment of the effectiveness of the 12-week comprehensive inpatient or outpatient rehabilitation treatment carried out as prevention against functional disability in BT patients with various degrees of disability after the end of oncological treatment. 

## 2. Materials and Methods 

### 2.1. Study Design

This is a case-series report on two programs and a prospective observational clinical study which was carried out between 2014 and 2019 in accordance with the principles of the Helsinki Declaration of the World Medical Association after obtaining consent from the Bioethics Committee of the Poznan University of Medical Sciences (No 12/2014). All patients participating in the study had to obtain approval from an oncologist and a neurosurgeon. In addition, a research physiatrist provided a detailed explanation of the study’s physical exercise program. 

### 2.2. Participants and Setting

As part of the study, we examined primary BT patients after surgery, radiotherapy, and/or chemotherapy up to a maximum of 30 days after the end of oncological treatment. Medically eligible patients received a study information letter from their physician. All study participants signed an informed consent about the study program and were enrolled according to the criteria. 

The inclusion criteria for rehabilitation treatment included: stable medical course, age between 18 and 75, and good general condition. Patients who had metastatic BTs, significant co-morbidities or were medically unstable (e.g., had cardiac diseases resulting in circulation failure above second class determined by the New York Heart Association (NYHA), uncontrolled asthma etc.), had concomitant neurodegenerative disease, or severe psychiatric issues (such as uncontrolled schizophrenia, being actively suicidal or physically aggressive—based on clinical judgement), or were in a persistent minimally conscious state or vegetative status as well as those BT patients who withdrew from the study before completing the 12 weeks of the rehabilitation program were excluded from the statistical analysis. The study flow chart is presented in Figure 1.

#### 2.2.1. Inpatient Group 

The inpatient group consisted of BT patients who were selected from those admitted to the Neurorehabilitation Unit in Bonifraterskie Health Center out of 102 individuals who were admitted to the neurorehabilitation ward following BT treatment. As many as 46 (45.10%) individuals were primarily excluded from the study: they did not meet the criteria of the study (23.91%), suffered from other neurological disorders (36.95%) or physical limitation (19.56%) that could influence the course of clinical trial or because they were not interested in participating in the research (10.8%), did not complete data about their cancer and treatment (6.52%), and were unable to give conscious written informed consent and/or communicate (2.17%). Therefore, 56 (54.90%) BT patients were accepted for the study. However, during 12 weeks of rehabilitation treatment, 27 (48.21%) patients withdrew from the research, including 15 (26.78%) subjects because of emotional problems (lack of motivation or non-stable emotional status such as depression, psychological, and/or physiological tiredness), 5 (8.9%) patients—due to progress in cancer disease that required further oncological or palliative therapy, and 7 individuals decided to leave the unit at their own request since they assessed that their functional state was satisfactory for them personally. Finally, 29 (28.43% of all individuals that were rehabilitated at the hospital) fully completed the rehabilitation program. The results from one patient were excluded from analysis due to lack of full data (Figure 1). 

#### 2.2.2. Outpatient Group

The outpatient group consisted of BT patients who were selected from patients who were treated in the Outpatient Ward in the Department of Rehabilitation in the Cancer Center. Out of 101 BT patients, 47 (46.53%) were primarily excluded from the research since they did not meet the study criteria (31.91%), had another neurological illness (23.4%), physical limitation (12.76%) that could influence the course of clinical trial (e.g., early after orthopedic surgery, lack of limb, severe hearing loss), had a problem with transportation to the center during 12 weeks of therapy (5.51%). Some patients were not interested in participating in the treatment (17.02%), or had emotional issues after oncological treatment and did not want to participate in our observation (6.38%). During the course of 12 weeks of outpatient rehabilitation, 54 (53.47%) patients participated in our program. However, 28 (51.85%) of participating BT patients withdrew from the research during therapy. Most of them (10 individuals—35.71%) resigned from the study or were not interested in participation since rehabilitation treatment was too long for them. As many as 11 (39.28%) subjects abandoned the research because of emotional problems, i.e., lack of motivation (28.57%) or non-stable emotional status such as depression (6.38%) or fatigue (6.38%). Moreover, 4 (8.5%) BT patients had progression in cancer disease and had to stop rehabilitation for oncological treatment. Finally, 26 (25.74% of all outpatient BT subjects who were rehabilitated) completed the rehabilitation program.

### 2.3. Rehabilitation Program

We observed BT patients in two rehabilitation programs: in a neurorehabilitation ward in the provincial rehabilitation center (as inpatient rehabilitation treatment) and during an outpatient BT rehabilitation program performed in a local cancer hospital. The patients were allocated to different rehabilitation treatments based on their independent self-care. If a patient needed any help in most daily activities (e.g., in body toilet, eating or transfers), he/she was enrolled in the inpatient ward program, but if the individual presented better self-physical functioning, he/she was allocated to the outpatient rehabilitation program.

#### 2.3.1. Inpatient Rehabilitation Program

All patients in the neurorehabilitation ward performed individual physical exercises 6 days/week for 150 min/day based on neuro-muscular re-education, including proprioceptive neuromuscular facilitation (PNF), Bobath therapy, coordination, balance and gait treatment, as well as endurance training after confirming the patient’s ability to participate independently in the exercise at 60–75% of their maximum heart rate (HR_max_ is age subtracted from 220). In addition, those patients had 1 h/day of individual occupational therapy, neuropsychological therapy, and speech therapy 5 days/week. Most participants followed the same diet which was baseline nourishment (for most of patients) or a diabetes mellitus diet.

#### 2.3.2. Outpatient Rehabilitation Program

BT patients enrolled in the outpatient rehabilitation program performed exercise training 5 days/week for 120 min/day (one hour of individual therapy with neuro-muscular reeducation exercises and one hour of group exercises using different devices such as cycling, treadmill, Pilates, yoga and general fitness exercises) at maximum 75% of HR_max_. In addition, patients had the opportunity to consult (individually or in group) a psychologist, a social worker, and an occupational therapist (5 days/per week). At each session, the patients had at least 3 h of therapy, 5 days per week.

### 2.4. Measurement 

The main outcome measure was a functional and cognitive assessment in both rehabilitation program groups following 12 weeks of comprehensive treatment. The second outcome was an analysis of changes in patient-reported scores of functional activities in BT patients. 

#### 2.4.1. Functional Assessment 

##### Functional Independence Measure

The Functional Independence Measure (FIM) is a measurement scale that explores an individual’s physical, psychological, and social functioning [15]. There are 18 items within the FIM. The motor section includes 13 items which assess the level of functioning in 4 subscales: self-care, transfers, locomotion, and sphincter control, whereas the cognition section comprises 5 items. Each item is rated on a scale of 1 to 7 (1—total assistance, 7—independence). The score shows a participant’s dependency in each area measured [15]. If the score falls below 6, another person is required for assistance or supervision. The FIM is extensively used to assess patients’ level of disability as well as changes in patient status in response to rehabilitation or medical intervention, also among patients with BT [15,16]. The tool is used to evaluate how well a person can carry out daily life activities and, thus, how dependent the patient would be on others [15]. By adding the points for each item, the possible total score ranges from 18 (lowest) to 126 (highest) levels of independence. During rehabilitation, admission, and discharge, scores are rated by a multidisciplinary rehabilitation team while observing patients’ functioning [15].

##### Barthel Index

The Barthel Index (BI) is a scale used to measure performance in activities of daily living (ADL) [17]. BI is a 10-item ordinal scale that estimates functional independence in the domains of patient care and mobility such as self-care, sphincter management, transfers, and locomotion [17,18]. It assesses levels of independence or dependence for 10 tasks with a score range of 0 (dependent) to 20 (independent) [17]. BI is one of the most widely used rating scales for the measurement of activity limitations in patients with neuromuscular and musculoskeletal conditions in a rehabilitation setting [18]. 

##### Berg Balance Scale 

The Berg Balance Scale (BBS) is a widely used clinical test assessing balance abilities in adults, used also among BT patients [19,20]. BBS aims to objectively determine patients’ ability (or inability) to safely balance during a series of tasks and can predict walking suitable for patients after rehabilitation therapy [19,20,21]. The qualitative tool assesses balance by means of functional tasks such as reaching, bending, standing, and transferring which involves most components of postural control [20]. Each item is scored using a 5-point scale (ranging from 0, which indicates the lowest level of function, to 4, denoting the highest level of function) with well-established criteria. The total score ranges from 0 to 56 [19].

#### 2.4.2. Cognitive Function Assessment

Cognitive function was evaluated using the Addenbrooke’s Cognitive Examination III (ACE III). The ACE III is a screening questionnaire that is capable of differentiating patients with and without cognitive impairment [22]. The ACE III consists of 21 questions, with a total score of 100. The results of each task are scored to give a total amount of 100 points (18 points for attention, 26 for memory, 14 for fluency, 26 for language, and 16 for visuospatial processing). The score needs to be interpreted in the context of a patient’s overall history and clinical examination [22,23]. The ACE III includes different scores for each function, in addition to the general score, to observe the progression of cognitive deficits over time [24].

#### 2.4.3. Patient Reported Outcome—Functional Assessment of Cancer Therapy 

The Functional Assessment of Cancer Therapy (FACT) scale has been commonly applied as a specific questionnaire measuring QoL in cancer patients as patient-reported outcome. This scale has been developed following principles of test construction and evaluation and has undergone psychometric testing for validity and reliability [25,26]. The Functional Assessment of Cancer Therapy-Brain (FACT-Br) and the Functional Assessment of Cancer Therapy-Cognitive Function (FACT-Cog) studies use questionnaires. 

##### The Functional Assessment of Cancer Therapy for Brain Tumors Scale (FACT-Br)

FACT-Br is a widely used specialized questionnaire to assess QoL with an additional brain subscale. It has previously been tested by Weitzner et al. [26] for validation and reliability. A total of 50 items are included in this questionnaire that cover the following domains of QoL: physical well-being, social/family well-being, emotional well-being, functional well-being, brain disorders, and other concerns [25,27]. Patients assess all items using a five-point Likert scale ranging from 0 “not at all” to 4 “very much”. The higher the ratings, the higher the QoL. 

##### The Functional Assessment of Cancer Therapy—Cognitive Scale (FACT-Cog) 

The FACT-Cog scale is designed to assess perceived cognitive function and its impact on QoL in cancer patients, and it is a validated measure expanded by Wagner et al. [28,29]. FACT-Cog assesses a range of self-reported cognitive functioning spheres in patients as perceived cognitive impairment or abilities, as well as their impact on QoL, and how the cognitive impairment is perceived by others, in addition to giving an overall cognitive function result, which is the sum of the four subscales [29]. Smaller values on this scale suggest greater cognitive difficulties and inefficiency.

### 2.5. Statistical Analysis

A priori, we calculated the sample size necessary to detect a significant, clinically important difference in outcome measurements over time (Group interaction effect, f statistic) between the inpatient and the outpatient BT group. Parameters of this calculation were: estimated effect size = 0.25, *p* < 0.05, power = 0.80, required sample: n = 50 (n = 25 per rehab group).

The normality of the distribution of the variables was tested using the Shapiro–Wilk test. A level of significance of 0.05 was used in the analysis. Thus, all *p*-values of less than 0.05 were interpreted as indicating significant correlations. The values of qualitative variables in the groups were compared using the chi-square test (with Yates’ correction) or the Fisher’s test for low expected values in the tables. The values of quantitative variables in the two groups were compared using the Student’s t-test (where the variable concerned had normal distribution in both groups) or the Mann–Whitney test (otherwise). The values of qualitative variables in the groups were compared using the chi-square test (with Yates’ correction for 2 × 2 tables) or the Fisher’s exact test for low expected values in the tables. The analysis was performed using R software, version 3.6.0. [30]. 

## 3. Results

### 3.1. Baseline Characteristics

The characteristics of the study groups on a baseline are provided in Table 1. The information includes socio-demographic data, comorbidities, BT lesion size and type, tumor grade, and general treatments received. Inpatients were older than outpatients (*p* < 0.05). Other characteristics did not differ significantly between the study groups (*p* > 0.05).

### 3.2. General Assessment at Admission to Rehabilitation Program

In the overall assessment, BT patients admitted to the two rehabilitation programs displayed statistically significant differences in clinical functional parameters (*p* < 0.05) before therapy, in the following scales: BI, BBS, FIM global, FIM sphincter control, FIM transfer and locomotion, and ACE visual-spatial functions. Most of the cognitive subscales of ACE III did not differ between the inpatient and outpatient participants (*p* > 0.05). Clinical assessment results at admission in both study groups are presented in Table 2.

### 3.3. Analysis of Physical Functioning Results in BT Inpatients and Outpatients

The analysis of motor functioning results (BI, BBS, and FIM) revealed statistically significant improvement in most parameters in inpatient and outpatient study participants alike, except for an insignificant improvement in the FIM social cognition subscale in the outpatient group. All physical functioning results obtained during the study are shown in Table 3.

### 3.4. Effect of Rehabilitation Treatment on Cognitive Functioning in BT Inpatients and Outpatients

The analysis of the cognitive function results indicated statistically significant improvement in most cognitive domains after 12 weeks of therapy in both inpatient and outpatient groups. Only in ACE III, the language functioning subscale did not show any significant change after the outpatient treatment program. The results of cognitive measurements in both BT rehabilitation groups are presented in Table 4.

### 3.5. Results of Rehabilitation Treatment in Patient-Reported Function Questionnaires

The analysis of the FACT questionnaire results demonstrated statistically significant improvement in all self-reported questions about functioning after 12 weeks of rehabilitation in the BT inpatients and outpatients alike, except for the FACT-Cog score of perceived cognitive abilities in outpatients. Results of the FACT questionnaire are shown in Table 5.

## 4. Discussion

Rehabilitation of BT individuals comprises of cancer rehabilitation that should be more widespread due to its importance for daily functioning in this group of cancer patients. Comprehensive rehabilitation treatment under the supervision of physiatrists with interdisciplinary collaboration among the medical staff (physiotherapist, psychologist, speech therapist, rehabilitation nurse, and social worker), with oncology-related assistance in managing side effects or symptoms (such as fatigue, sleep disturbance, depression, or seizure) or other medical complications is an important element of medical care for prevention of disabilities or impairment in BT patients [6,9,13]. While the traditional measures of clinical oncological care tend to focus on tumor-related outcomes (morbidity, survival time, histopathology, imaging, and side effects of oncological treatments), cancer rehabilitation programs and outcome measurements offer a more functional approach that focuses on the total impact of treatments on the patient’s life and social participation [13]. The present study demonstrates the results of 12-weeks therapy in two types of supervised rehabilitation programs which were performed on BT patients according to their functional status. It was noticed that the long-lasting rehabilitation treatment plays a significant role in observing clinically useful improvement in many aspects of functioning in BT patients despite significant differences in physical functioning and oncological treatment before BT outpatients and inpatients started the rehabilitation program. As a result of rehabilitation care, we were able to observe improvements especially in the motor functioning results (such as BI, BBS, and FIM) and statistically significant improvement in most parameters in both study groups. Despite the remarkable results of the two different rehabilitation treatment methods, we observed a significant number of patients who did not complete the program which was its limitation. Moreover, we observed statistically and clinically significant improvement in daily functioning in both study groups.

In accordance with the principles of the International Classification of Functioning, Disability and Health (ICF) for improving patients’ life, comprehensive rehabilitation treatment could consist of categories of health domains described from the perspective of Body Functions and Structures; and Activities and Participation [31]. Several clinical studies have evaluated functional outcomes for BT patients from a rehabilitation perspective [11,12,32,33,34,35] but are methodologically imperfect (bias, lack of control group, and blinding) [6]. BT patients have a huge potential to improve from comprehensive rehabilitation, and the intensity, timing, and pace of activity should be provided in accordance with patients’ goals and status [5,6].

The present study showed two types of rehabilitation program in the early phase once a BT treatment is completed (to maximum one month after completion of oncological treatment). The goal of the research was to study the inpatient or outpatient rehabilitation treatment effectiveness in decreasing disability and functional impairment which had been caused by oncotherapy in BT patients with neurological deficits. This clinical study provides an answer to the question whether the 12-week rehabilitation treatment under an inpatient or outpatient program provides sufficient prevention against functional disability to facilitate BT patients’ self-management ability after oncological treatment. Patients in the two types of rehabilitation programs were enrolled according to the study criteria and functional independence (when patients required help in most everyday activities and could not reach the rehabilitation hospital with their families, they were admitted to the inpatient rehabilitation center). Patients with BT have varying degrees of functional and psychological impairment because of factors relating to the tumor or to the treatment they received [6,7,8]. The study patients with statistically significant impairment manifested a decrease in physical functioning measurements such as BI (below 14 points) or BBS (under 40 points), FIM (below 90 points) and their subscales, and without social cognition and communication were enrolled in the inpatient center. Early exercise therapy was introduced because of the potential to modify many biochemical factors and to improve brain function after cancer treatment [32]. In this study, all patients first performed individual physical exercises and cognitive training adapted to the needs of BT patients and their current capabilities.

The study used BI, FIM, and additionally BBS to assess physical functioning, as these scales had previously been applied with positive recommendations in BT patients [14,16,19,21,34] to assess problems reported by the participants or observed by the physiatrist, and which were linked to the ICF categories [36,37]. The FIM scale is a daily living activity scoring system that may objectively determine impairments in different domains [15,37]. The analysis of the results in many domains of physical functioning indicated that both inpatient and outpatient 12-week rehabilitation programs are effective in improving the general physical condition of BT patients.

As previous studies have shown [7,10,38,39,40], cognitive rehabilitation is strongly connected with physical training and rehabilitation using neuro-muscular types of exercises. The use of physical activity, neurocognitive training, behavioral therapy to treat memory and mood, and removal of drugs that may be associated with neurocognitive side effects (e.g., anti-epileptic drugs) helped BT patients to manage the neurocognitive impairments [39,40]. BTs can cause global cognitive dysfunction by disruption of cognitive networks, with memory, attention, or executive functioning being the most frequently affected domains [7,41]. Cognitive assessment allows rehabilitation team members to set goals for rehabilitation treatment and facilitates decision-making about further intervention and monitoring its progress [5,7,13]. In the present study, the cognition levels did not differ significantly between the two groups but indicated important deficits in many cognitive domains before rehabilitation treatment in both these groups. ACE III was used as a well-tolerated screening score which permits differentiation of patients with cognitive impairment, including those with BTs [22,23,42]. This questionnaire is a more sensitive cognition tool than the mini-mental state examination (MMSE) in BT patients. For example, as Brown et al. [43] observed, only few patients with low-grade gliomas screened with MMSE had cognitive deterioration which is very often impaired in BT patients after radiation therapy. ACE III, like other neuropsychological tests (such as MMSE and MoCA), provides the psychologist with a quick and global cognitive screen of the patient, specifying both measures of each domain and the overall cognitive profile [44]. In the present study, BT patients had significant cognitive dysfunctions of memory, attention, orientation, and visual spatial functions. Implementation of the individual cognitive therapy in the rehabilitation program (especially as a daily cognitive-related training in the inpatient group) resulted in substantial improvement in many cognitive domains without pharmacological support. On a physiological level, some researches [45,46] underline regular exercise training as a way to potentially increase the levels of serotonin, brain-derived neurotrophic factor (BDNF), and vascular endothelial growth factor (VEGF), to improve cognitive functioning. Other authors [38,47] highlighted regular physical activity as an important factor for the improvement of patients’ mood, decreasing the symptoms of anxiety, depression, or chronic pain. Therefore, limited cognitive function in BT patients can have a cumulative effect over time and lower QoL. The type of rehabilitation program to be applied should be strongly connected with patients’ general functioning (cognitive and physical) after cancer treatment, so a patient’s self-reported measure is needed for the comprehensive clinical assessment [28,48].

The National Institute for Health and Clinical Excellence [49] has advised using the QoL assessment to measure participation in daily function as a self-reported functional measure in BT subjects [50]. In this study, the FACT-Br and additionally FACT-Cog questionnaires have been applied to measure self-related cognitive functioning. A patient-reported outcome is a health outcome directly reported by the patient who experienced it and is being increasingly incorporated into oncology research [51] in contrast to an outcome reported by other medical staff. The analysis of our results using the FACT questionnaires confirmed an improvement in daily functioning in BT patients after the inpatient and outpatient rehabilitation program alike. Only the Perceived Cognitive Abilities domain in the FACT-Cog scale did not change significantly after the outpatient rehabilitation program.

Within the ICF, BT-related impairments can limit activity or function and participation in daily life, and reduce QoL [37,52]. Multidisciplinary rehabilitation defined as the coordinated multidimensional intervention provided by medical professionals improves patient functional independence and participation using a holistic biopsychosocial model, as defined by the ICF, and might reduce disability in BT patients after oncological treatment.

Needless to say, this research has some limitations. Despite the large number of BT patients in rehabilitation wards, we enrolled only slightly above 25% of subjects to this study. Most of the individuals were excluded according to study criteria, including some patients who were not interested in taking part in this study program. Numerous patients, both in the outpatient and inpatient groups, withdrew from the study due to neuro-psychiatric illnesses and psychological complications. It is well known that BT patients suffer from many psychological disorders such as depression, fatigue syndrome, and emotional instability which disturb their daily functioning [7,11,41,42]. Our participants tolerated the prescribed exercises well but reported emotional problems despite the regular psychological care during rehabilitation. Some of the patients decided to leave the unit at their own request since they assessed their functional state as satisfactory. Therefore, they wanted to go home as soon as possible despite encouragement to continue the rehabilitation program because of its positive effect on their functioning. Those groups of patients had a feeling of fear and uncertainty as well as the desire to leave hospital at the earliest. Due to the extended study time, some patients had no motivation to continue the program and were satisfied with their functioning before the end of the study. Some of the individuals had disease progression. It must be emphasized that these patients had a heterogeneous disease course and, therefore, improvements are not likely to be universal simply by participation in a rehabilitation program. Thus, we noticed that rehabilitation time should be determined in consultation with patients as it may be influenced by their motivation.

On the other hand, the research has some strengths. Two rehabilitation programs (in- and out-patient) that achieved good effects in both groups of patients with various degrees of disabilities were presented. It should be emphasized that the individuals were properly selected to the study with the proper sample size necessary to detect a significant, clinically important difference in outcome measurements over time. Moreover, different scales were used to evaluate comprehensively the functional status of the patients.

To sum up, one may say that BT patients need to have a comprehensive and intensive rehabilitation program introduced as soon as possible after oncological treatment, including not only physical exercises, but also support from a psychologist, social worker, and occupational therapist. Such sessions should be individually arranged depending on the functional status of the patient. However, a good effect is achieved, if the activities are systematic and they are carried out 5 days a week. In this way, the general status of the individual might be improved, i.e., functional, mental, and emotional state, and such effects may finally result in a good QoL for BT patients.

## 5. Conclusions

The 12-week rehabilitation which started early after BT treatment as an inpatient or outpatient program had a positive role in improving functions in BT patients in all aspects of their functioning in terms of physical function, subjective neurocognition, and QoL. The authors of this trial recommend comprehensive rehabilitation treatment to be implemented as soon as possible once oncological therapy is completed, for the purpose of preventing disabilities in this group of cancer patients.

## Figures and Tables

**Figure 1 ijerph-17-06488-f001:**
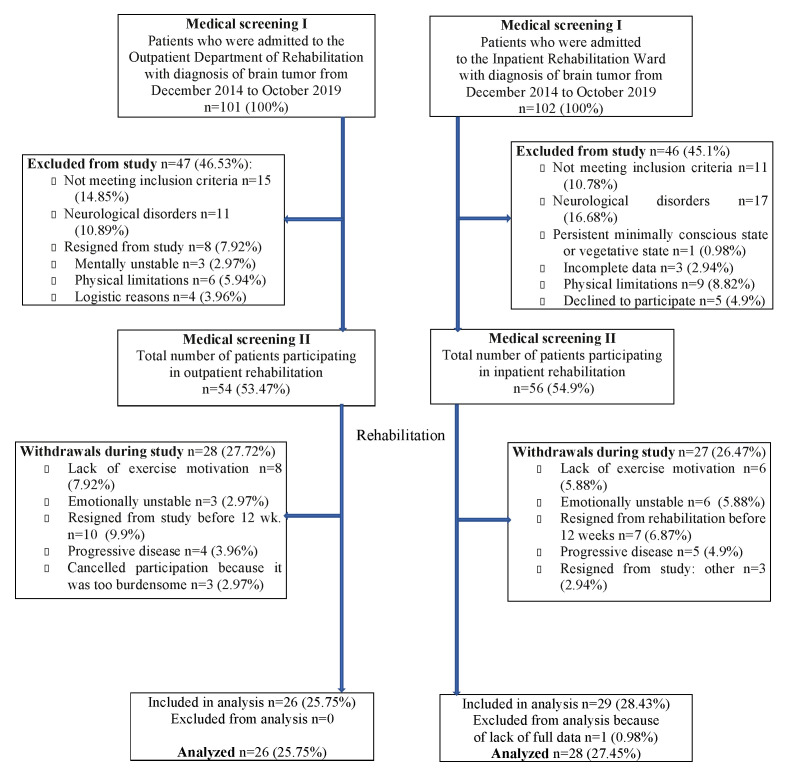
Flow of participants (as outpatients and inpatients) throughout the study.

**Table 1 ijerph-17-06488-t001:** Characteristics of the study groups.

Parameters		Inpatient (*n* = 28)	Outpatient (*n* = 26)	Both Groups (*n* = 54)	*p*-Value
Age	Mean ± SD	54.39 ±13.66	47.08 ± 7.77	50.87 ± 11.71	0.02
	Median	55	47	49	*p*
	QR	43.25–65.25	43.25–53	43.25–58.25	
Sex	Female	14 (50.00%)	11 (42.31%)	25 (46.30%)	0.769
	Male	14 (50.00%)	15 (57.69%)	29 (53.70%)	chi ^2^
WHO grades of selected CNS tumors according to the 2016 CNS WHO [3]	Diffuse astrocytic and oligodendroglial tumors				0.716 F
	diffuse astrocytoma II	3 (10.71%)	2 (7.69%)	5 (9.26%)	
	oligodendrogliglioma IDH-mutant II	1 (3.57%)	0 (0.00%)	1 (1.85%)	
	anaplastic astrocytoma, IDH-mutant III	2 (7.14%)	0 (0.00%)	2 (3.70%)	
	glioblastoma, IDH-wildtype IV	3 (10.71%)	6 (23.08%)	9 (16.67%)	
	diffuse midline glioma, H3K27M-mutant IV	1 (3.57%)	3 (11.54%)	4 (7.41%)	
	Other astrocytic tumors				
	pilocytic astrocytoma I	0 (0.00%)	1 (3.85%)	1 (1.85%)	
	subependymal giant cell astrocytoma I	4 (14.29%)	2 (7.69%)	6 (11.11%)	
	Ependymal tumors				
	subependynoma I	1 (3.57%)	1 (3.85%)	2 (3.70%)	
	ependynoma I/II	0 (0.00%)	1 (3.85%)	1 (1,85%)	
	anaplastic ependynoma III	1 (3.57%)	0 (0.00%)	1 (1.85%)	
	Meningiomas				
	meningioma I	7 (25.00%)	7 (26.92%)	14 (25.93%)	
	atypical meningioma II	5 (17.86%)	3 (11.54%)	8 (14.81%)	
Side of paresis	Left	14 (50.00%)	15 (57.69%)	29 (53.70%)	0.763
	Right	11 (39.29%)	10 (38.46%)	21 (38.89%)	F
	Both sides	3 (10.71%)	1 (3.85%)	4 (7.41%)	
Radiotherapy	No	16 (57.14%)	13 (50.00%)	29 (53.70%)	0.8
	Yes	12 (42.86%)	13 (50.00%)	25 (46.30%)	chi ^2^
Chemotherapy	No	24 (85.71%)	17 (65.38%)	41 (75.93%)	0.153
	Yes	4 (14.29%)	9 (34.62%)	13 (24.07%)	chi ^2^

chi ^2^ = chi-square test, F = Fisher test; *p* = parametrical analysis using Student’s T-test.

**Table 2 ijerph-17-06488-t002:** Characteristics of functional status of BT patient groups before rehabilitation treatment.

Parameters	Inpatient (*n* = 28) Median; QR	Outpatient (*n* = 26) Median; QR	*p*-Value
BI	9; 5.75–13	16; 14–16.75	<0.001
BBS	29; 13.5–38	34.5; 30–42	0.007
FIM total	77; 49.25–89.75	91; 84–100.75	0.001
FIM self-care	26; 17–30.5	31; 27.25–35	0.005
FIM sphincter control	10; 6.75–12.5	12; 10.25–14	0.019 *
FIM transfers/stability	13; 6–16	14; 12–16	0.012 *
FIM locomotion	4.5; 0–10	10; 8–12	0.002 *
FIM communication	10; 6.75–12	10; 10–12	0.159 *
FIM social cognition	15; 8–17	15; 10–16.75	0.727 *
ACE III global	74.5; 53–80	77.5; 70.25–80.75	0.212 *
ACE III attention and orientation	14; 9.75–17	16; 14–17.75	0.085 *
ACE III memory	17.5; 10–22	19; 15–22	0.361 *
ACE III language fluency	9; 4.75–10	10; 7.25–11.75	0.255 *
ACE III language functioning	20.5; 12.25–24.25	21.5; 19.25–24.75	0.313 *
ACE III visual-spatial functions	9; 7.75–13.25	11; 9–14	0.043
MMSE	22; 19–28.75	23; 20–29.5	0.144 *

*p* = paired T-test, *p* * = paired Wilcoxon test (due to lack of normality).

**Table 3 ijerph-17-06488-t003:** Analysis of the physical functioning results before and after rehabilitation treatment in inpatient and outpatient brain tumor patient groups.

Parameters	Rehabilitation Treatment	*N*	Baseline Mean ± SD	After 12 Weeks Mean ± SD	*p*-Value
BI score	IP	28	9.07 ± 4.35	14.46 ± 4.76	<0.001
	OP	26	15.27 ± 2.13	17.73 ± 1.76	<0.001
BBS score	IP	28	25.96 ± 14.52	40.18 ± 12.52	<0.001
	OP	26	35.08 ± 8.05	43.42 ± 7.69	<0.001 *
FIM Total score	IP	28	71.29 ± 29.94	95.89 ± 29.14	<0.001
	OP	26	93.23 ± 13.15	107.35 ± 10.84	<0.001
FIM self-care subscale	IP	28	24.57 ± 9.96	31.93 ± 9.67	<0.001
	OP	26	31 ± 5.07	36.12 ± 4.57	<0.001
FIM sphincter control subscale	IP	28	9.29 ± 4.42	11.39 ± 4.32	0.005 *
	OP	26	12.12 ± 3.19	14.35 ± 3.1	<0.001
FIM transfers subscale	IP	28	11.36 ± 6.02	15.75 ± 4.76	<0.001
	OP	26	14.69 ± 2.62	17.38 ± 2.65	<0.001 *
FIM locomotion subscale	IP	28	5.36 ± 5.04	10.18 ± 4.04	<0.001
	OP	26	9.69 ± 2.72	11.85 ± 1.93	<0.001 *
FIM communication subscale	IP	28	9.29 ± 4.17	11.29 ± 3.95	0.006 *
	OP	26	11.27 ± 2.54	12.62 ± 2.06	<0.001 *
FIM social cognition subscale	IP	28	12.71 ± 5.67	17.25 ± 5.15	<0.001 *
	OP	26	13.77 ± 3.98	14.5 ± 3.36	0.18

*p* = paired T-test, *p* * = paired Wilcoxon test (due to lack of normality).

**Table 4 ijerph-17-06488-t004:** Analysis of the cognitive functioning results before and after rehabilitation treatment in BT inpatients and outpatients.

Parameters	Rehabilitation Program	*N*	Baseline Mean ± SD	After 12 Weeks Mean ± SD	*p*-Value
ACE III global	IP	28	65.5 ± 23.77	76.79 ± 23.78	<0.001
	OP	26	75.58 ± 11.99	82.19 ± 11.23	<0.001
ACE III attention and orientation	IP	28	12.89 ± 4.75	15.07 ± 4.66	0.001
	OP	26	15.35 ± 2.91	16.88 ± 2.7	0.001
ACE III memory	IP	28	15.75 ± 6.9	18.93 ± 7.13	0.001
	OP	26	17.96 ± 4.85	19.42 ± 4.75	0.002
ACE III language fluency	IP	28	8 ± 3.69	9.79 ± 4.21	0.002 *
	OP	26	9.19 ± 3.07	11.12 ± 2.7	<0.001 *
ACE III language functioning	IP	28	17.57 ± 8.35	20.14 ± 7.22	0.001
	OP	26	21.23 ± 3.66	21.92 ± 3.85	0.059 *
ACE III visual-spatial functions	IP	28	9.21 ± 4.61	11.43 ± 4.73	<0.001
	OP	26	11.5 ± 3.35	12.69 ± 3.02	0.006 *

*p* = paired T-test, *p* * = paired Wilcoxon test (due to lack of normality).

**Table 5 ijerph-17-06488-t005:** Analysis of the patient-reported function questionnaire results before and after rehabilitation treatment in BT inpatients and outpatients.

Scores	Rehabilitation Program	*N*	Baseline Mean ± SD	After 12 Weeks Mean ± SD	*p*-Value
FACT-Brain Total	IP	28	89.93 ± 6.31	113.82 ± 6.13	<0.001
	OP	26	99.31 ± 5.92	129.31 ± 5.18	<0.001
FACT-General Total	IP	28	45.07 ± 4.74	56.64 ± 4.68	<0.001
	OP	26	54.23 ± 3.25	66.27 ± 4.07	<0.001
FACT–Physical Well-Being	IP	28	11.5 ± 3.37	18.43 ± 2.59	<0.001
	OP	26	14.5 ± 2.28	21.35 ± 2.33	<0.001
FACT–Social/Family Well-Being	IP	28	12.21 ± 1.91	10.18 ± 1.83	<0.001
	OP	26	14.58 ± 1.68	12.85 ± 1.91	0.003
FACT- Emotional Well-Being	IP	28	9.93 ± 2.23	14.82 ± 1.56	<0.001
	OP	26	11.19 ± 2.19	14.46 ± 1.56	<0.001
FACT–Functional Well-Being	IP	28	11.43 ± 2.15	13.21 ± 2.22	0.001
	OP	26	13.96 ± 2.11	17.62 ± 2.04	<0.001
FACT–Brain Cancer	IP	28	44.86 ± 3.03	57.18 ± 3.45	<0.001
	OP	26	45.08 ± 4.15	63.04 ± 4.3	<0.001
FACT-Cognitive Total	IP	28	48.61 ± 5.57	82.57 ± 7.08	<0.001
	OP	26	64.23 ± 2.42	89 ± 4.09	<0.001
FACT-Perceived Cognitive Impairments	IP	28	27.32 ± 5.58	45.07 ± 5.54	<0.001
	OP	26	34.92 ± 3.08	53.73 ± 3.4	<0.001 *
FACT-Impact of Perceived Cognitive Impairments on QoL	IP	28	3.86 ± 0.89	9.21 ± 1.6	<0.001
	OP	26	5.19 ± 0.94	8.5 ± 1.33	<0.001 *
FACT-Comments from Others	IP	28	6.29 ± 1.96	9.93 ± 1.7	<0.001 *
	OP	26	7.19 ± 1.67	9.27 ± 1.69	<0.001
FACT-Perceived Cognitive Abilities	IP	28	11.14 ± 1.74	18.36 ± 2.66	<0.001
	OP	26	16.92 ± 2.23	17.5 ± 1.86	0.248

*p* = paired T-test, *p* * = paired Wilcoxon test (due to lack of normality).

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
