# Peer review of "Can Multidisciplinary Inpatient and Outpatient Rehabilitation Provide Sufficient Prevention of Disability in Patients with a Brain Tumor?—A Case-Series Report of Two Programs and A Prospective, Observational Clinical Trial"

_ijerph, 2020, doi:10.3390/ijerph17186488_

Round 1
Reviewer 1 Report
The authors present a well written paper regarding the effectiveness of both outpatient and inpatient rehabilitation in patients with primary brain tumor. I have a few minor comments regarding wording. Table 1 also needs to be fixed.
1) Line 14. Please change to: "Brain tumor patients have a high incidence of disability due to the effects of the tumor itself or oncological treatment. Despite the neurological and functional deficits caused by..."
2) Line 23: Please change to: "Analysis of the results showed that patients in both programs achieved significant improvement in physical functioning scores in activities of daily living"
3) Line 33: Please change to: "Globally, a total of 296,851 new cases..."
4) Line 41: Please change to: "...neurological impairments and restore quality of life (QoL) [5,6]."
5) Line 64: Please change to: "Other authors [6,14] have noted a lac kof recommendations for rehabilitation management in BT patients."
6) Line 75: Please change "neurosurgeons" to "neurosurgeon".
7) Line 78: Please change to: "In this study, we examined primary BT patients after surgery..."
8) Line 124: Please change to: "There are 18 items within the FIM..."
9) Line 258: Please change to: "Comprehensive rehabilitation treatment under the supervision..."
10) Line 348: Please change to: "The 12-week rehabilitation which started early after BT treatment as an inpatient and outpatient program alike has a positive role in improving function in BT patients..."
11) Table 1: "Type of cancer" appears to have some problems.
a) Ependymoma III is misspelled.
b) The word "glioma" is often used interchangeably with astrocytoma. I suspect the "glioma I" is really a WHO Grade I Astrocytoma. The "glioma IV" is probably a WHO Grade IV Astrocytoma. Please discuss with a pathologist at your institution to clarify the type of cancer. A WHO Grade IV Astrocytoma is a glioblastoma multiforme.
The glioblastoma III might be an WHO Grade III Astrocytoma and the glioblastoma II may be a WHO Grade II Astrocytoma.
Author Response
We would like to thank you for your suggestions. We have made our point-to-point changes below. Please take a look at our corrections.
Comment 1 1) Line 14. Please change to: "Brain tumor patients have a high incidence of disability due to the effects of the tumor itself or oncological treatment. Despite the neurological and functional deficits caused by..." Response Thank you very much, you are right – we have made the correction. Comment 2 2) Line 23: Please change to: "Analysis of the results showed that patients in both programs achieved significant improvement in physical functioning scores in activities of daily living" Response This information has been corrected, thank you. Comment 3 3) Line 33: Please change to: "Globally, a total of 296,851 new cases..." Response This information has been corrected, thank you Comment 4 4) Line 41: Please change to: "...neurological impairments and restore quality of life (QoL) [5,6]." Response Thank you, you are right – we have made the correction. Comment 5 5) Line 64: Please change to: "Other authors [6,14] have noted a lot of recommendations for rehabilitation management in BT patients." Response This information has been changed, thank you Comment 6 6) Line 75: Please change "neurosurgeons" to "neurosurgeon". Response This mistake has been corrected, thank you. Comment 7 7) Line 78: Please change to: "In this study, we examined primary BT patients after surgery..." Response Thank you, we have made the change. Comment 8 8) Line 124: Please change to: "There are 18 items within the FIM..." Response Thank you, you are right – we have made the correction. Comment 9 9) Line 258: Please change to: "Comprehensive rehabilitation treatment under the supervision..." Response Thank you, we have changed it. Comment 10 10) Line 348: Please change to: "The 12-week rehabilitation which started early after BT treatment as an inpatient and outpatient program alike has a positive role in improving function in BT patients..." Response We have made the correction. Thank you. |
Comment 11 Table 1: "Type of cancer" appears to have some problems. Response a/ The mistake has been corrected, thank you. b) The word "glioma" is often used interchangeably with astrocytoma. I suspect the "glioma I" is really a WHO Grade I Astrocytoma. The "glioma IV" is probably a WHO Grade IV Astrocytoma. Please discuss with a pathologist at your institution to clarify the type of cancer. A WHO Grade IV Astrocytoma is a glioblastoma multiforme. The glioblastoma III might be an WHO Grade III Astrocytoma and the glioblastoma II may be a WHO Grade II Astrocytoma. Response b/ We absolutely agree with this statement. Thank you for the suggestions concerning discussion with a pathologist at our cancer center. Finally, we have decided to present the type of brain tumors in our patients according to the 2016 World Health Organization classification of tumors of the central nervous system which was presented in Acta Neuropatholog. 2016, 131, 803–820. [doi:10.1007/s00401-016-1545-1]. We hope that this way of describing the type of brain tumor in our study will satisfy you. |
Reviewer 2 Report
This is an excellent report of the potential benefits of both inpatient and outpatient rehabilitation in patients experiencing disability from brain tumors and their treatment. The study has numerous strengths including rigorous methodology and a comprehensive QOL panel that spans both physical and cognitive/other domains, and is very well written. Its value is in providing objective data supporting the notion that both inpatient and outpatient rehabilitation lead to objective improvements in many of the recorded domains.
However, though the authors do convincingly show that patients who complete the rehab regimen demonstrate improvements from baseline, more discussion should be devoted to the limitations of the study. First, only 30% of patients were assessable in both inpatient and outpatient groups (see Figure 1), attesting to the fact that these are patients with a heterogeneous disease course and therefore improvements are likely not universal or "guaranteed" by participation in a rehab program. Second, the authors compare outpatient vs. inpatient programs at several point (eg Results section 3.5), and this is not an appropriate comparison given the fact that patients selected for inpatient programs are by definition a completely group of patients than those selected for outpatient programs, as the authors themselves acknowledge. Thus, I do not feel that the comparisons between outpatient vs. inpatient programs are useful, and the paper would be better served by presenting inpatient vs. outpatient outcomes completely independently (which the authors do in most sections of the paper).
Also, one small error I believe is contained in Table 2: FIM sphincter control, for outpatient group there appears to be a range displayed (10.25-14) but not the median.
Author Response
Thank you very much for the nice opinion. We have made our point-to-point replies below. Please take a look at our comments:
Comment 1 However, though the authors do convincingly show that patients who complete the rehab regimen demonstrate improvements from baseline, more discussion should be devoted to the limitations of the study. First, only 30% of patients were assessable in both inpatient and outpatient groups (see Figure 1), attesting to the fact that these are patients with a heterogeneous disease course and therefore improvements are likely not universal or "guaranteed" by participation in a rehab program. Response Thank you. I have changed discussion. We hope that we have not missed anything and explained well in discussion the reasons for a significant reduction in the number of patients who rehab programs completed. Comment 2 Second, the authors compare outpatient vs. inpatient programs at several point (eg Results section 3.5), and this is not an appropriate comparison given the fact that patients selected for inpatient programs are by definition a completely group of patients than those selected for outpatient programs, as the authors themselves acknowledge. Thus, I do not feel that the comparisons between outpatient vs. inpatient programs are useful, and the paper would be better served by presenting inpatient vs. outpatient outcomes completely independently (which the authors do in most sections of the paper). Response I would like to thank you for your suggestions. We have removed the part of text where there was comparison between outpatient vs. inpatient programs at several point (in Results section 3.5). Comment 3 Also, one small error I believe is contained in Table 2: FIM sphincter control, for outpatient group there appears to be a range displayed (10.25-14) but not the median. Response The missing data has been completed, thank you.
|
Reviewer 3 Report
Evidence in support of rehabilitation following diagnosis of brain tumour is lacking and as such, the findings from this work provide novel contributions to the wider evidence base. My queries following review are as follows:
- authors need to justify the use of labeling the study as a 'controlled clinical trial'. Does the rehab program form part of standard practice and if so, this is more likely a case-series report of two programs. If not standard practice, then this paper presents pre- and post-findings from 2 rehab programs. It is unclear why the authors are interested in comparing the effect of the two rehab programs when a) eligibility is different for the two - the inpatient required patients to need assistance with daily activities, and b) the rehab programs were different. Instead, it does seem relevant and important to present what changes occurred in outcomes measured as a result of participating in one of the two rehab programs.
- Improvements to the introduction is needed including ensuring writing style is scientific (avoid use of emotive terms such as 'suffer'), place comments in context (e.g., are statistics given worldwide, specific to a country?) and there needs to be a better synthesis of the current evidence base. That is, what do we know about rehab for people with BT and what are the limitations around this knowledge. In turn, this information can be used to justify this study.
- In line with my query about study design, presentation of results that compare the two groups should be removed. The focus should be on changes observed within each of the groups. Authors also need to comment on sample size calculations (that is, is the analyses adequately-powered) and most importantly, need to comment on whether findings are clinically relevant (this is more relevant than statistical significance). further, given approx 50% of the sample in both groups failed to contribute data, authors need to compare characteristics of completers vs non-completers and comment on the likely bias of the results.
- Would highly recommend a reorientation of the discussion so that it first deals with the findings from this work (that is, participating in rehab was associated with improvements in....), followed by how these findings extend current understanding, strengths and limitations of the study and the clinical implications of the findings.
Author Response
We would like to thank you for your suggestions. Please take a look at our comments below:
Comment 1 Authors need to justify the use of labeling the study as a 'controlled clinical trial'. Does the rehab program form part of standard practice and if so, this is more likely a case-series report of two programs. If not standard practice, then this paper presents pre- and post-findings from 2 rehab programs. It is unclear why the authors are interested in comparing the effect of the two rehab programs when a) eligibility is different for the two - the inpatient required patients to need assistance with daily activities, and b) the rehab programs were different. Instead, it does seem relevant and important to present what changes occurred in outcomes measured as a result of participating in one of the two rehab programs. Response Thank you very much for this suggestion, you are completely right. We have made a correction concerning study design. We hope, we have not missed anything and that this section now explains well the subject discussed. Currently, despite of many recommendations, the rehab program in brain tumor patients is not a standard practice in the world. Only in selected regions of countries some hospitals conduct brain tumor rehab for this group of patients. Therefore, in our study we wanted to show our results of two different rehab programs which were clinically useful and good tolerated by brain tumor patients. At the beginning of the study we assumed that it could be a prospective and observation trial because we did not know the possible outcomes of our rehab programs. Comment 2 Improvements to the introduction is needed including ensuring writing style is scientific (avoid use of emotive terms such as 'suffer'), place comments in context (e.g., are statistics given worldwide, specific to a country?) and there needs to be a better synthesis of the current evidence base. That is, what do we know about rehab for people with BT and what are the limitations around this knowledge. In turn, this information can be used to justify this study. Response Thank you. The introduction has been heavily reedited according to your comments. We hope we have not missed anything. We also hope that this section explains well the discussed subjects. Comment 3 In line with my query about study design, presentation of results that compare the two groups should be removed. The focus should be on changes observed within each of the groups. Authors also need to comment on sample size calculations (that is, is the analyses adequately-powered) and most importantly, need to comment on whether findings are clinically relevant (this is more relevant than statistical significance). further, given approx 50% of the sample in both groups failed to contribute data, authors need to compare characteristics of completers vs non-completers and comment on the likely bias of the results. Response Thank you for the suggestions concerning the comparison of two groups. We have removed this part of study from the manuscript. Moreover, according to your suggestion, we have added information about sample size calculations and the data about clinically useful results. Additionally, we have tried our best to describe properly the characteristics of completers vs non-completers of the study, and comment on the likely bias of the results. Thank you for this comment. Comment 4 Would highly recommend a reorientation of the discussion so that it first deals with the findings from this work (that is, participating in rehab was associated with improvements in....), followed by how these findings extend current understanding, strengths and limitations of the study and the clinical implications of the findings. Response We have completely rewritten the discussion according to your recommendation. Please, check if it is sufficient and satisfactory. The relevant term has been explained and the part about the information.
|